

# SECAPR—a bioinformatics pipeline for the rapid and user-friendly processing of targeted enriched Illumina sequences, from raw reads to alignments

Tobias Andermann[1,2], Ángela Cano[2,3], Alexander Zizka[1,2], Christine Bacon[1,2] and Alexandre Antonelli[1,2,4,5]

[1] Department of Biological and Environmental Sciences, University of Gothenburg, Gothenburg, Sweden
[2] Gothenburg Global Biodiversity Centre, Gothenburg, Sweden
[3] Department of Botany and Plant Biology, University of Geneva, Geneva, Switzerland
[4] Gothenburg Botanical Garden, Gothenburg, Sweden
[5] Department of Organismic and Evolutionary Biology, Harvard University, Cambridge, MA, United States of America

Corresponding author
Tobias Andermann,
tobias.andermann@bioenv.gu.se

## ABSTRACT

Evolutionary biology has entered an era of unprecedented amounts of DNA sequence data, as new sequencing technologies such as Massive Parallel Sequencing (MPS) can generate billions of nucleotides within less than a day. The current bottleneck is how to efficiently handle, process, and analyze such large amounts of data in an automated and reproducible way. To tackle these challenges we introduce the Sequence Capture Processor (SECAPR) pipeline for processing raw sequencing data into multiple sequence alignments for downstream phylogenetic and phylogeographic analyses. SECAPR is user-friendly and we provide an exhaustive empirical data tutorial intended for users with no prior experience with analyzing MPS output. SECAPR is particularly useful for the processing of sequence capture (synonyms: target or hybrid enrichment) datasets for non-model organisms, as we demonstrate using an empirical sequence capture dataset of the palm genus *Geonoma* (Arecaceae). Various quality control and plotting functions help the user to decide on the most suitable settings for even challenging datasets. SECAPR is an easy-to-use, free, and versatile pipeline, aimed to enable efficient and reproducible processing of MPS data for many samples in parallel.

## INTRODUCTION

An increasing number of studies apply sequence data generated by Massive Parallel Sequencing (MPS) to answer phylogeographic and phylogenetic questions (e.g., *Botero-Castro et al., 2013*; *Smith et al., 2014a*; *Smith et al., 2014b*; *Faircloth et al., 2015*; *Heyduk et al., 2016*). Researchers often decide to selectively enrich and sequence specific genomic regions of interest, rather than sequencing the complete genome. One reason is that enriching specific markers leads to a higher sequencing depth for each individual marker, as compared to the alternative of sequencing full genomes. Sequencing depth is important

for the extraction of single nucleotide polymorphisms (SNPs) and for allele phasing (*Andermann et al., 2018*; *Bravo et al., 2018*). Additionally, phylogenetic analysis software usually relies on multiple sequence alignments (MSAs) with homologous sequences across many taxa, which are simple to recover when specifically enriching these sequences prior to sequencing.

The enrichment of specific genomic regions (markers) is usually achieved through sequence capture (synonyms: hybrid enrichment, hybrid selection, exon capture, target capture) prior to sequencing (*Gnirke et al., 2009*). This technique applies specific RNA baits, which hybridize with the target regions and can be captured with magnetic beads. Sequence capture is gaining popularity as more bait sets for non-model organisms are being developed. Some bait sets are designed to match one specific taxonomic group (e.g., *Heyduk et al., 2016*; *Kadlec et al., 2017*), while others are designed to function as more universal markers to capture homologous sequences across broad groups of taxa (e.g., UCEs, *Faircloth et al., 2012*). After enrichment of targeted markers with such bait sets, the enriched sequence libraries are sequenced on a MPS machine (see *Reuter, Spacek & Snyder, 2015*).

Despite recent technological developments, analyzing sequencing results is a great challenge due to the amount of data produced by MPS machines. An average dataset often contains dozens to hundreds of samples, each with up to millions of sequencing reads. Such amounts of sequence data require advanced bioinformatics skills for storing, quality checking, and processing the data, which may represent an obstacle for many students and researchers. This bottleneck calls for streamlined, integrative and user-friendly pipeline solutions.

To tackle these challenges, here we introduce the Sequence Capture Processor (SECAPR) pipeline, a semi-automated workflow to guide users from raw sequencing results to cleaned and filtered multiple sequence alignments (MSAs) for phylogenetic and phylogeographic analyses. We designed many of the functionalities of this pipeline toward sequence capture datasets in particular, but it can be effectively applied to any MPS dataset generated with Illumina sequencing (Illumina Inc., San Diego, CA, USA). SECAPR comes with a detailed documentation in form of an empirical data tutorial, which is explicitly written to guide users with no previous experience with MPS datasets. To simplify the processing of big datasets, all available functions are built to process batches of samples, rather than individual files. We developed SECAPR to provide the maximum amount of automation, while at the same time allowing the user to choose appropriate settings for their specific datasets. The pipeline provides several plotting and quality-control functions, as well as more advanced processing options such as the assembly of fully phased allele sequences for diploid organisms (*Andermann et al., 2018*).

## MATERIAL & METHODS

### The SECAPR pipeline in a nutshell

SECAPR is a platform-independent pipeline written in python, and tested for full functionality on Linux and MacOS. It can be easily downloaded together with all

its dependencies as a virtual environment, using the conda package manager (see 'Availability'). The strength of SECAPR is that it channels the main functionalities of many commonly used bioinformatics programs and enables the user to apply these to sets of samples, rather than having to apply different software to each sample individually. In addition, SECAPR is optimized for high performance computing as it enables computational parallelization for all major functions, which allows the efficient processing of datasets encompassing dozens to hundreds of samples in parallel.

The basic SECAPR workflow (Fig. 1) includes the following steps:

1. *Quality filtering and adapter trimming*;
2. *De novo contig assembly*;
3. *Selection of target contigs*;
4. *Building MSAs from contigs*;
5. *Reference-based assembly*;
6. *Allele phasing.*

SECAPR automatically writes summary statistics for each processing step and sample to a log-file (*summary_stats.txt*, Table 1). The pipeline includes multiple visualization options to gauge data quality and, if necessary, adapt processing settings accordingly. SECAPR comes with a detailed documentation and data tutorial (see 'Availability').

## Description of the SECAPR workflow

*1. Quality filtering and adapter trimming (secapr clean_reads).* The SECAPR *clean_reads* function applies the software Trimmomatic (*Bolger, Lohse & Usadel, 2014*) for removing adapter contamination and low quality sequences from the raw sequencing reads (FASTQ-format). An additional SECAPR plotting function summarizes FASTQC (*BabrahamBioinformatics*) quality reports of all files and produces a visual overview of the whole dataset (Fig. 2). This helps to gauge if the files are sufficiently cleaned or if the *clean_reads* function should be rerun with different settings.

*2. De novo contig assembly (secapr assemble_reads).* The SECAPR function *assemble_reads* assembles overlapping FASTQ reads into longer sequences (*de novo* contigs) by implementing the *de novo* assembly software Abyss (*Simpson et al., 2009*). Abyss has been identified as one of the best-performing DNA sequence assemblers currently available (*Hunt et al., 2014*). As an alternative to Abyss we also implemented the option to use the Trinity assembler (*Grabherr et al., 2011*), which currently is only supported for the Linux distribution of SECAPR. We do however recommend the use of Abyss as the preferred assembly-software, firstly due to significantly faster computation and secondly and more importantly due to the fact that Trinity is intended for the assembly of RNA transcriptome data, leading to different assumptions about the input data in comparison to DNA assemblers such as Abyss, as discussed in *Haas et al. (2013)*.

*3. Selection of target contigs (secapr find_target_contigs).* The SECAPR function *find_target_contigs* identifies and extracts those contigs that represent the DNA targets of interest. This function implements the program LASTZ (formerly BLASTZ, *Harris, 2007*) by searching the contig files for matches with a user-provided FASTA-formatted reference library. For sequence capture datasets, a suitable reference library is the reference

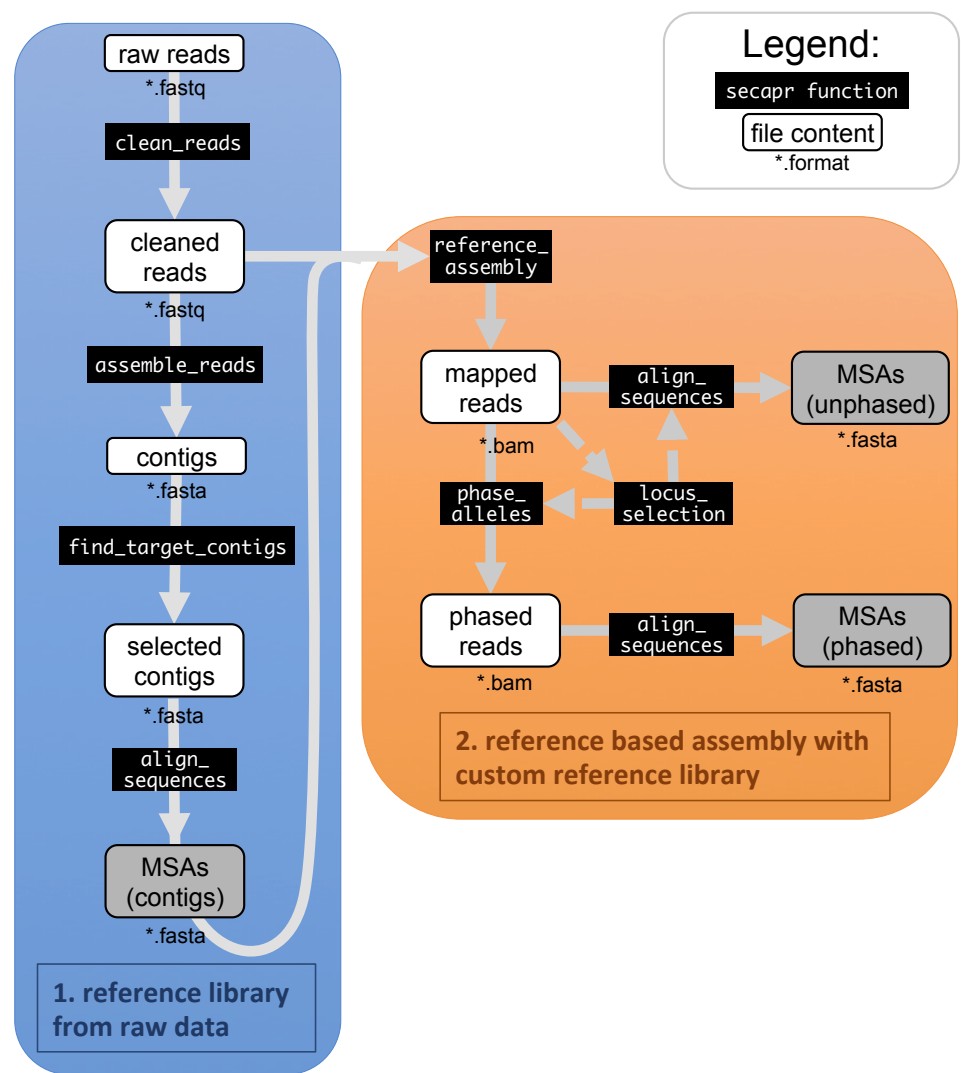

**Figure 1** **SECAPR analytical workflow.** The flowchart shows the basic SECAPR functions, which are separated into two steps (colored boxes). Blue box (1. reference library from raw data): in this step the raw reads are cleaned and assembled into contigs (*de novo* assembly); orange box (2. reference based assembly with custom reference library): the contigs from the previous step are used for reference-based assembly, enabling allele phasing and additional quality control options, e.g., concerning read-coverage. Black boxes show SECAPR commands and white boxes represent the input and output data of the respective function. Boxes marked in grey represent multiple sequence alignments (MSAs) generated with SECAPR, which can be used for phylogenetic inference.

file that was used for synthesizing the RNA baits, which will return all contigs that match the enriched loci of interest. The *find_target_contigs* function identifies potentially paralogous loci (loci that have several matching contigs) and excludes these from further processing. It further allows the user to keep or exclude long contigs that match several adjacent reference loci, which can occur if the reference file contains sequences that are located in close proximity to each other on the genome (e.g., several separate exons of the same gene).

**Table 1** **Summary statistics for all samples, produced by SECAPR.** Reported for each sample (1. column) are the number of sequencing reads in the FASTQ sequencing files, before (2. column) and after (3. column) cleaning and trimming, the total count of assembled *de novo* contigs (4. column), the number of filtered contigs that matched target loci (5. column) and the number of sequencing reads that mapped to the new reference library generated from the contig MSAs during reference-based assembly (6. column). These summary statistics are automatically compiled and appended to a log file (*summary_stats.txt*) during different steps in the SECAPR pipeline.

| Sample ID | FASTQ read pairs (raw) | FASTQ read pairs (cleaned) | Total contig count | Recovered target contigs | Reads on target regions |
|---|---|---|---|---|---|
| 1087 | 291,089 | 276,072 | 277,628 | 562 | 22,308 |
| 1086 | 244,726 | 231,326 | 230,122 | 516 | 17,969 |
| 1140 | 206,106 | 192,676 | 153,377 | 469 | 18,039 |
| 1083 | 377,228 | 352,646 | 309,993 | 534 | 31,922 |
| 1082 | 277,999 | 262,378 | 258,359 | 556 | 19,491 |
| 1085 | 307,671 | 291,377 | 309,561 | 512 | 22,030 |
| 1079 | 315,801 | 298,450 | 306,369 | 550 | 13,969 |
| 1061 | 209,586 | 192,407 | 177,910 | 545 | 14,474 |
| 1068 | 295,402 | 278,069 | 264,865 | 563 | 22,013 |
| 1063 | 354,795 | 336,356 | 356,512 | 525 | 20,439 |
| 1080 | 459,485 | 434,951 | 433,954 | 531 | 41,068 |
| 1065 | 217,725 | 205,290 | 204,082 | 544 | 13,524 |
| 1073 | 302,798 | 286,021 | 289,612 | 529 | 15,598 |
| 1070 | 295,822 | 278,011 | 295,557 | 539 | 19,288 |
| 1064 | 408,723 | 384,908 | 405,080 | 543 | 21,531 |
| 1074 | 408,370 | 383,604 | 398,758 | 531 | 25,476 |
| 1166 | 405,667 | 385,442 | 410,292 | 544 | 29,697 |

*4. Building MSAs from contigs (secapr align_sequences).* The SECAPR function *align_sequences* builds multiple sequence alignments (MSAs) from the target contigs that were identified in the previous step. The function builds a separate MSA for each locus with matching contigs for ≥3 samples.

*5. Reference-based assembly (secapr reference_assembly).* The SECAPR *reference_assembly* function applies the BWA mapper (*Li & Durbin, 2010*) for reference-based assembly of FASTQ reads and Picard (broadinstitute.github.io/picard/) for removing duplicate reads. The function saves the assembly results as BAM files (Fig. 3), transforms them into Variant Call Format (VCF) format using SAMtools (*Li et al., 2009*), and generates a consensus sequence from the read variation at each locus. These consensus sequences have several advantages over the *de novo* contig sequences (see 'Discussion') and can be used for building MSAs with the SECAPR *align_sequences* function.

The *reference_assembly* function includes different options for generating a reference library for all loci of interest:

- *−reference_type alignment-consensus*: The user provides a link to a folder containing MSAs, e.g., the folder with the contig MSAs from the previous step, and the function calculates a consensus sequence from each alignment. These consensus sequences are then used as the reference sequence for the assembly. This function is recommended

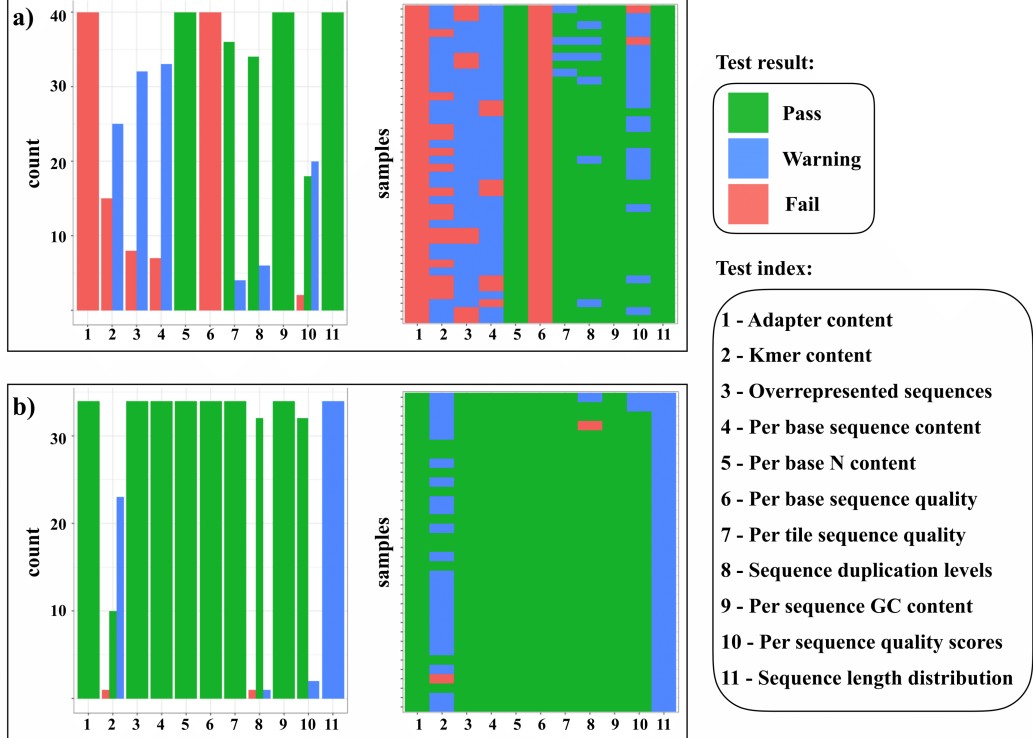

**Figure 2  Overview of FASTQc quality test result.** (A) Before and (B) after cleaning and adapter trimming of sequencing reads with the SECAPR function *clean_reads*. This plot, as produced by SECAPR, provides an overview of the complete dataset and helps to gauge if the chosen cleaning parameters are appropriate for the dataset. The summary plots show the FASTQc test results, divided into three categories: passed (green), warning (blue) and failed (red). The *x*-axis of all plots contains the eleven different quality tests (see legend). The bar-plots ('count') represent the counts of each test result (pass, warning or fail) across all samples. The matrix plots ('samples') show the test result of each test for each sample individually (*y*-axis). This information can be used to evaluate both, which specific parameters need to be adjusted and which samples are the most problematic.

when running reference-based assembly for groups of closely related samples (e.g., samples from the same genus or family).

- *–reference_type sample-specific*: From the MSAs, the function extracts the contig sequences for each sample and uses them as a sample-specific reference library. If the user decides to use this function it is recommended to only use alignments for reference that contain sequences for all samples. This will ensure that the same loci are being assembled for all samples.

- *–reference_type user-ref-lib*: The user can provide a FASTA file containing a custom reference library.

An additional SECAPR function (*locus_selection*) allows the user to select a subset of the data consisting of only those loci, which have the best read-coverage across all samples.

*6. Allele phasing (secapr phase_alleles).* The SECAPR *phase_alleles* function can be used to sort out the two phases (reads covering different alleles) at a given locus. This function
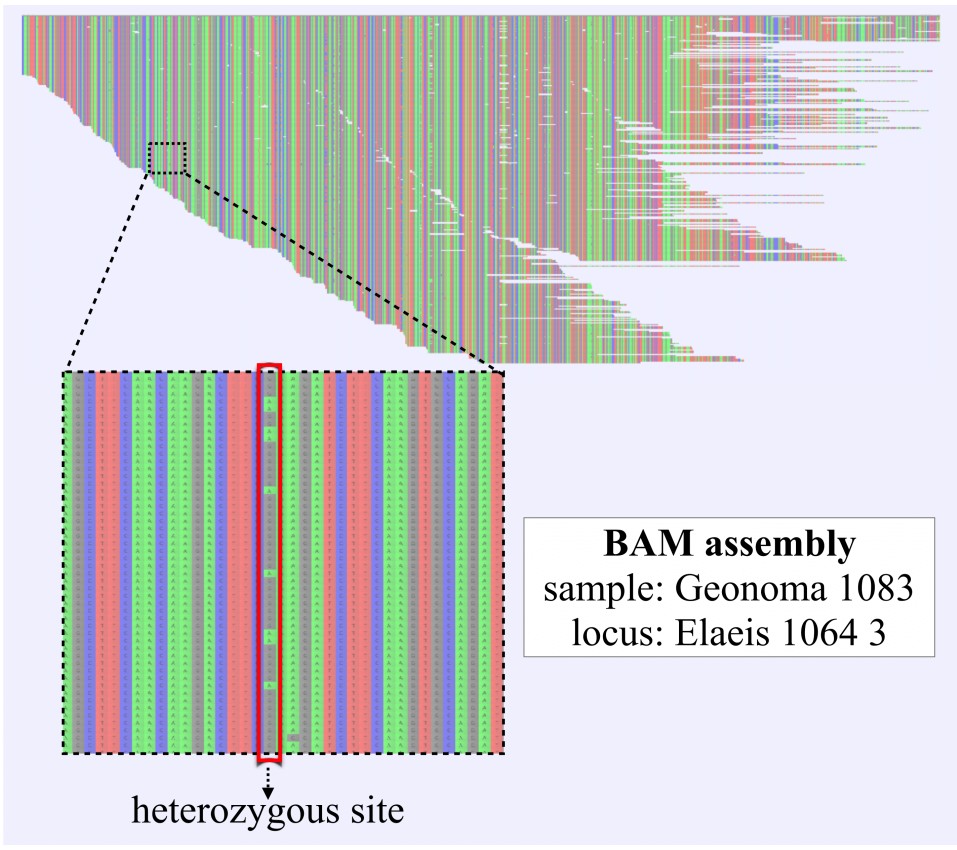

**Figure 3 Reference-based assembly including heterozygous sites.** BAM-assembly file as generated with the SECAPR *reference_assembly* function, shown exemplarily for one exon locus (1/837) of one of the *Geonoma* samples (1/17). The displayed assembly contains all FASTQ sequencing reads that could be mapped to the reference sequence. The reference sequence in this case is the *de-novo* contig that was matched to the reference exon 'Elaeis 1064 3'. DNA bases are color-coded (A, green; C, blue; G, black; T, red). The enlarged section contains a heterozygous site, which likely represents allelic variation, as both variants A and G are found at approximately equal ratio.

applies the phasing algorithm as implemented in SAMtools, which uses read connectivity across multiple variable sites to determine the two phases of any given diploid locus (*He et al., 2010*). After running the phasing algorithm, the *phase_alleles* function outputs a separate BAM-file for each allele and generates consensus sequences from these allele BAM-files. This results into two sequences at each locus for each sample, all of which are collected in one cumulative sequence file (FASTA). This sequence file can be run through the SECAPR *align_sequences* function in order to produce MSAs of allele sequences.

7. Pipeline automation (*secapr automate_all*). SECAPR provides the *automate_all* function that produces contig MSAs and phased allele MSAs from cleaned FASTQ files, only requiring one single command, automating steps 2–6. The user can choose between three different settings, namely *relaxed*, *medium* or *conservative*, which run the SECAPR pipeline with different sensitivity parameters. The *relaxed* setting is recommended when samples in the dataset are expected to show considerable genetic variation (e.g., samples from
different families) or are expected to differ considerably from the provided reference library used for the extraction of target contigs. The *conservative* setting on the other hand uses very restrictive similarity thresholds for the identification of target contigs and during reference-based assembly and is therefore recommended for datasets containing closely related samples (within same genus) that are expected to also be similar to the provided reference library. The *medium* setting constitutes an intermediate between these two extremes and is recommendable for datasets, which contain samples of closely related genera. While in all cases it is recommendable to run the individual SECAPR functions with customized settings for each specific dataset, the *automate_all* function can help to get a first impression of the dataset through the SECAPR-logged sample information (Table 1) and the inspection of the produced MSAs and other, intermediate files, such as the reference-assembly BAM-files.

## Benchmarking with empirical data

We demonstrate the functionalities of SECAPR on a novel dataset of target sequencing reads of *Geonoma*, one of the most species-rich palm genera (plant family Arecaceae) of tropical Central and South America (*Dransfield et al., 2008*; *Henderson, 2011*). Our data comprise newly generated Illumina sequence data for 17 samples of 14 *Geonoma* species (Table S1), enriched through sequence capture. The bait set for sequence capture was designed specifically for palms by *Heyduk et al. (2016)* to target 176 genes with in total 837 exons. More detailed information about the generation of the sequence data can be found in Appendix S1. All settings and commands used during processing of the sequence data can be found in the SECAPR documentation on our GitHub page (see 'Availability'). An example of a downstream application of the MSAs produced by SECAPR for phylogeny estimation can be found in Appendix S2.

## RESULTS

The newly generated *Geonoma* data used for benchmarking constitute an empirical example of a challenging dataset, characterized by irregular read coverage and multiple haplotypes. Despite these challenges, the SECAPR workflow provides the user all the necessary functions to filter and process datasets into MSAs for downstream phylogenetic analyses.

After *de novo* assembly (*secapr assemble_reads*) we recovered an average of 535 (stdev = 22) contigs per sample (*secapr find_target_contigs*) that matched sequences of the 837 targeted exons (Table 1, Fig. 4A, Table S2). In total 120 exons were recovered for all samples. Many of the recovered target contigs spanned several reference exons (all samples: mean = 100, stdev = 25) and hence were flagged as contigs matching multiple loci (Table S3). Since these contigs may be phylogenetically valuable, as they contain the highly variable interspersed introns, we decided to keep these sequences. We extracted these longer contigs together with all other non-duplicated contigs that matched the reference library (*secapr find_target_contigs*) and generated MSAs for each locus that could be recovered in at least three *Geonoma* samples (*secapr align_sequences*). This resulted in contig alignments for 692 exon loci (Fig. 4A).

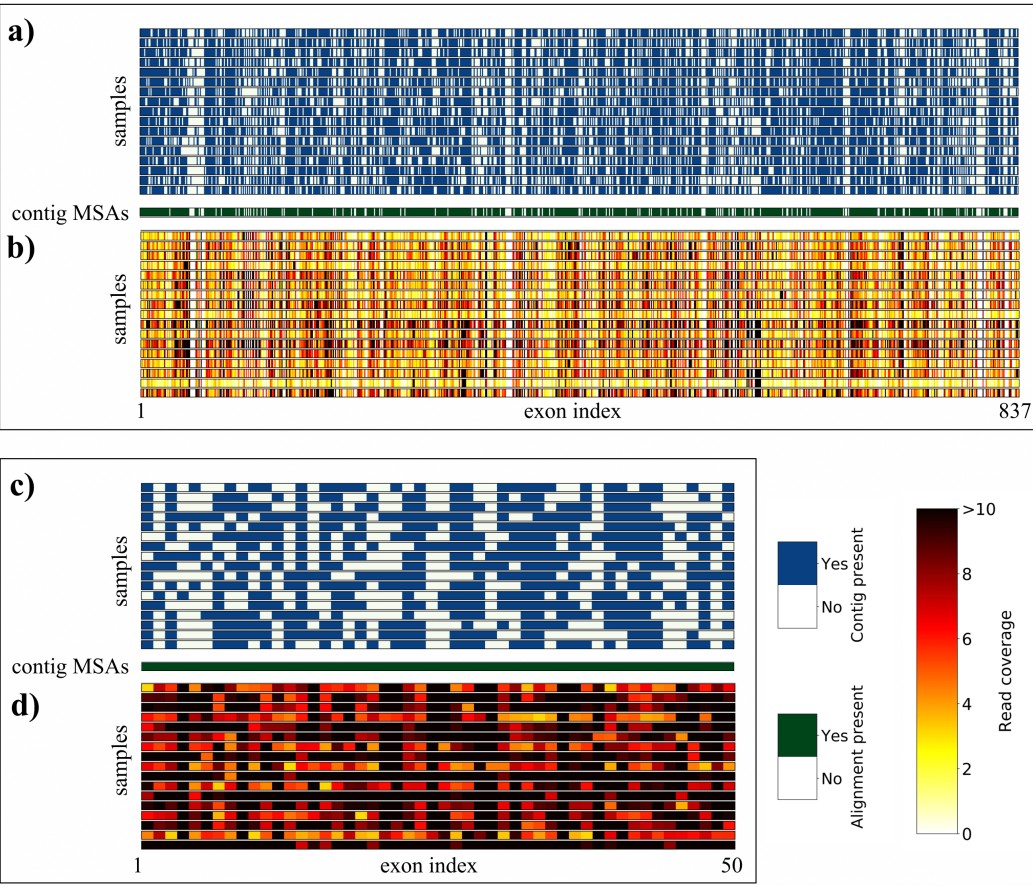

**Figure 4** **Overview of sequence yield for *Geonoma* sample data, produced with SECAPR.** The matrix plots show an overview of the contig yield and read-coverage for all targeted loci (A + B) and for the selection of the 50 loci with the best read coverage (C + D), selected with the SECAPR function *locus_selection* (see Table S5 for loci-names corresponding to indices on *x*-axes). (A) and (C) show if *de novo* contigs could be assembled (blue) or not (white) for the respective locus (column) and sample (row). Contig MSAs were generated for all loci that could be recovered for at least three samples (green). (B) and (D) show the read coverage (see legend) for each exon locus after reference-based assembly. The reference library for the assembly consisted of the consensus sequences of each contig MSA, and hence is genus specific for *Geonoma*.

During reference-based assembly (*secapr reference_assembly*) we mapped the reads against the consensus sequence of the contig MSAs for all loci. We found an average of 454 exon loci (stdev = 108) per sample that were covered by more than three reads (average coverage across complete locus, Fig. 4B). A total of 67 exon loci had an average read coverage of more than three reads across all samples (Table S4). We extracted the 50 loci with the best coverage across all samples (*secapr locus_selection*), as shown in Figs. 4C and 4D. In cases of irregular read-coverage across samples (as in our sample *Geonoma* data), we strongly recommend the use of the *locus_selection* function before further processing the data, as demonstrated in our tutorial (see 'Availability').

The results of the reference-based assembly also revealed that our sample data showed more than two haplotypes for many loci. Future research may clarify whether this is the result of various paralogous loci in the dataset or if our *Geonoma* samples are polyploid due to a recent genome duplication or hybridization event in the ancestry of the genus. Due to the presence of more than two haplotypes at various loci, the results of the allele-phasing step (*secapr phase_alleles*) are to be viewed critically, since the algorithm is built for phasing the read data of diploid organisms or loci only. Hence, the allele-phasing results are not reported or discussed in the scope of this manuscript. All phased BAM files and the compiled allele MSAs are available online (see 'Availability').

## DISCUSSION

### *De novo* assembly vs. reference-based assembly

There are several ways of generating full sequences from raw FASTQ-formatted sequencing reads. The SECAPR pipeline contains two different approaches, namely *de novo* assembly and reference-based assembly (Fig. 1). *De novo* assembly can be directly applied to any raw read data while reference-based assembly requires the user to provide reference sequences for the assembly. We find for the *Geonoma* example data that reference-based assembly results into recovering the majority of all target sequences per sample (Fig. 4D) and provides the user a better handle on quality and coverage thresholds. It is also computationally much less demanding in comparison to *de novo* assembly.

However, reference-based assembly is very sensitive toward the user providing orthologous reference sequences that are similar enough to the sequencing reads of the studied organisms. If the reference sequences are too divergent from the sequenced organisms, only a small fraction of the existing orthologous sequencing reads will be successfully assembled for each locus. In contrast, when relaxing similarity thresholds and other mapping parameters too much (e.g., to increase the fraction of reads included in the assembly) there is a higher risk of assembling non-orthologous reads, which can lead to chimeric sequences being assembled. This can be a problem, particularly in cases of datasets containing non-model organisms, since suitable reference sequences for all loci usually do not exist.

For this reason, the SECAPR workflow encourages the user to use these two different assembly approaches in concert (Fig. 1). Our general suggestion is to first assemble contig MSAs for all regions of interest, resulting from *de novo* assembly and then use these MSAs to build a reference library for reference-based assembly. In that case SECAPR produces a reference library from the sequencing data itself, which is specific for the taxonomic group of interest or even for the individual sample.

A common approach is to stop data processing after the *de novo* assembly step and then use the contig MSAs for phylogenetic analyses (e.g., *Faircloth et al., 2012*; *Smith et al., 2014a*; *Smith et al., 2014b*; *Faircloth, 2015*). Here we take additional processing steps, including generating new reference libraries for all samples and using these for reference-based assembly. There may be several reasons for carrying out these additional steps:

1. Sensitivity: In order to identify *de novo* contigs that are orthologous to the loci of interest, the user is usually forced (because of the lack of availability) to use a set of

reference sequences for many or all loci that are not derived from the studied group. Additionally these reference sequences may be more similar to some sequenced samples than to others, which can introduce a bias in that the number of recovered target loci per sample is based on how divergent their sequences are to the reference sequence library. In other words, the 'one size fits all' approach for recovering contig sequences is not the preferred option for most datasets and may lead to taxonomic biases. For this reason it is recommended to generate family, genus or even sample-specific reference libraries using the recovered contigs, and use these to re-assemble the sequencing reads.

2. Intron/exon structure: The advantage of creating a new reference library from the contig data, rather than using alternatively available sequences as reference (e.g., the RNA bait sequences) for reference-based assembly, is that available reference sequences often constitute exons, omitting the interspersed intron sequences (as in the case of bait sequences). The more variable introns in between exons are usually not suitable for designing baits, they are too variable, but are extremely useful for most phylogenetic analyses because they have more parsimony informative sites. There is a good chance that the assembled contigs will contain parts of the trailing introns or even span across the complete intron, connecting two exon sequences (e.g., *Bi et al., 2012*). This is why it is preferable to use these usually longer and more complete contig sequences for reference-based assembly, rather than the shorter exon sequences from the bait sequence file, in order to capture all reads that match either the exon or the trailing intron sequences at a locus.

3. Allelic variation: Remapping the reads in the process of reference-based assembly will identify the different allele sequences at a given locus. This can also aid in the evaluation of the ploidy level of samples and in identifying loci potentially affected by paralogy.

4. Coverage: Reference-based assembly will give the user a better and more intuitive overview over read-depth for all loci. There are excellent visualization softwares (such as Tablet, by *Milne et al., 2013*) that help interpret the results.

## Novelty

Several pipelines and collections of bioinformatics tools exist for processing sequencing reads generated by MPS techniques, e.g., PHYLUCE (*Faircloth, 2015*), GATK (*McKenna et al., 2010*) and 'reads2trees' (*Heyduk et al., 2016*). In contrast to some of these existing pipelines, SECAPR (i) is targeted towards assembling full sequence data (as compared to only SNP data, e.g., GATK); (ii) is intended for general use (rather than project specific, e.g., reads2trees); (iii) is optimized particularly for non-model organisms and non-standardized sequence capture datasets (as compared to specific exon sets, e.g., PHYLUCE); (iv) allows allele phasing and selection of the best loci based on read coverage, which to our knowledge are novel to SECAPR. This is possible due to the approach of generating a clade- or even sample-specific reference library from the sequencing read data, which is then used for reference-based assembly; (v) offers new tools and plotting functions to give the user an overview of the sequencing data after each processing step.

## CONCLUSIONS

The SECAPR pipeline described here constitutes a bioinformatic tool for the processing and alignment of raw Illumina sequence data. It is particularly useful for sequence capture datasets and we show here how it can be applied to even challenging datasets of non-model organisms.

## AVAILABILITY

The SECAPR pipeline is open source and freely available, and can be found together with installation instructions, a detailed documentation and an empirical data tutorial at http://www.github.com/AntonelliLab/seqcap_processor.

## ACKNOWLEDGEMENTS

We thank Goo Jun, Corinne Grover and one anonymous reviewer for valuable feedback on earlier drafts of this manuscript. Further, we thank Estelle Proux-Wéra and Marcel Martin at the National Bioinformatics Infrastructure Sweden at SciLifeLab for their support with turning the SECAPR pipeline into a functioning conda package and for additional support in software development questions. The code for some of the functions of the SECAPR pipeline is inspired from similar functions in the PHYLUCE pipeline (*Faircloth, 2015*).

### Funding

This work was supported by the Swedish Research Council (B0569601), the European Research Council under the European Union's Seventh Framework Programme (FP/2007-2013, ERC Grant Agreement n. 331024), the Swedish Foundation for Strategic Research, the Faculty of Science at the University of Gothenburg, the David Rockefeller Center for Latin American Studies at Harvard University, and a Wallenberg Academy Fellowship to Alexandre Antonelli; and a SciLifeLab Bioinformatics Long-term Support from the Wallenberg Advanced Bioinformatics Infrastructure to Alexandre Antonelli and Bengt Oxelman. The funders had no role in study design, data collection and analysis, decision to publish, or preparation of the manuscript.

### Grant Disclosures

The following grant information was disclosed by the authors:
Swedish Research Council: B0569601.
European Research Council: FP/2007-2013, ERC Grant Agreement n. 331024.
Swedish Foundation for Strategic Research.
Faculty of Science at the University of Gothenburg.
David Rockefeller Center for Latin American Studies at Harvard University.
Wallenberg Academy Fellowship to Alexandre Antonelli.
SciLifeLab Bioinformatics Long-term.

## Competing Interests

The authors declare there are no competing interests.

## Author Contributions

- Tobias Andermann conceived and designed the experiments, analyzed the data, contributed reagents/materials/analysis tools, prepared figures and/or tables, authored or reviewed drafts of the paper, approved the final draft.
- Ángela Cano and Alexander Zizka contributed reagents/materials/analysis tools, prepared figures and/or tables, authored or reviewed drafts of the paper, approved the final draft, reviewed the manuscript.
- Christine Bacon and Alexandre Antonelli conceived and designed the experiments, authored or reviewed drafts of the paper, approved the final draft, reviewed the manuscript.

## DNA Deposition

The following information was supplied regarding the deposition of DNA sequences:

The raw FASTQ sequences of the Geonoma sample data are available at the NCBI Short Read Archive (SRA) via accession number SRP131660.

## Data Availability

Zenodo: 10.5281/zenodo.1162653.

## Supplemental Information

Supplemental information for this article can be found online at http://dx.doi.org/10.7717/peerj.5175#supplemental-information.

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
