# Peer review of "SECAPR—a bioinformatics pipeline for the rapid and user-friendly processing of targeted enriched Illumina sequences, from raw reads to alignments"

_PeerJ, doi:10.7717/peerj.5175_

## Round 0.1 · original submission · Minor Revisions

Please refer to the detailed reviewers comments regarding the manuscript and the pipeline, especially regarding the errors in running the pipeline.

Reviewer 1 ·

Basic reporting

The authors present a generally well written article that details a bioinformatic tool for the alignment/assembly and consensus calling of targeted sequence data.

The article is generally clear, unambiguous and well referenced.

I have some minor suggestions for improvement below:

Title – suggest adding “targeted enriched” to the title to discriminate this pipeline from those designed to work on whole-genome data.

Line 51 –“ which are easiest to recover when specifically enriching these sequences across all samples prior to sequencing”
Suggest changing to “simple to recover”. It is easy to extract homologous sequences for phylogenies regardless of pre-sequencing enrichment.

Line 51: “archived” should be changed to “achieved”

Line 106: Is FASTQC referenced properly here?

Line 110 to 115: Is Trinity implemented as well? Would be good to include if so.

Line 135: Using SAMTools?

Line 148: “Recommendable” should be “recommended”

Line 204-205: Wouldn’t this result be due to polyploidy within the Geonoma genus? Could the authors comment on this?

Experimental design

The software tool was well designed (installation was very simple, thank-you!) and easy to use. Dependencies are handled well, which is important for a tool that relies on so much other software.

Although it doesn't affect the outcome of the study it would have been nice if the authors included more in the way of pipeline automation. Working my way through the tutorial was easy and everything was well explained but the major point of a pipeline is to automate the vast majority of steps and only require user input at critical stages. Surely not all stages of the pipeline would be critical for every data-set? Is there an easy way that someone without a good bioinformatic skill-set be able to automate the majority of the pipeline?

Some consideration with regard to using this pipeline on high performance computing infrastructure would be useful. For example running this pipeline on 1000 samples would be very difficult without writing additional resource management scripts to help submit jobs to a scheduler. Granted this is probably out of the real of most researchers but these resources are becoming increasingly common and SECAPR is designed to be largely automated and able to process large numbers of samples.

Some additional information about downstream approaches to phylogenetic analyses would help improve the paper (and online tutorial). After such a detailed and thorough guide through the workflow a novice user would be left with assembled contigs and alignment files but have difficulties with the next steps.

Validity of the findings

The data and data analysis are well presented in the article.

·

Basic reporting

no comment

Experimental design

Here the authors developed a pipeline for assembling and using targeted sequence capture in non-model organisms, which has been a clear deficit in the current bioinformatic arena. The pipeline seems more or less sound; however, I wanted to use the pipeline on my own data to gauge success when paralogs are expected; when the distance between reference bait sequences and the recovered sequence varies; and where polyploidy is expected for some samples.

I encountered a minimum of two problems, which both delayed this review and caused me to stop testing. The first error was in a section of the code for assemble_reads; i.e., the region in the python script which calls ABySS was commented out. This error actually took a little while to resolve since my first instinct was the the conda package was not playing well with our spack module environment. Assembly was successfully completed after removing these comment marks.

The second error may also be due to a conflict in the conda evironment versus our existing spack module enviroment; however, I cannot be certain that there are no other errors preventing the code from running without spending more time on this.

I would be happy to provide a bug report on the developer's github; however, it may be just as quick for the authors to bundle this for spack (https://spack.readthedocs.io/en/latest/index.html). I am not sure the relative prevalence of spack versus conda versus other module systems; however, porting this to spack should be relatively easy and make it more broadly accessible as spack gains in popularity.

Validity of the findings

no comment

---

## Round 0.2 · accepted · Accept

As an editor, I am satisfied with the revision and I believe all the reviewers' comments are properly reflected in the revision. Please note that PeerJ does not provide proof editing, so make sure that everything on your manuscript looks correct.

#